# A Safe Frost Growth Screening Method to Accurately Predict Ice Plug Formation Completion during Waterpipe Freezing Repairs

**Fei-Lung Liu** [1,*] **, Shu-Kai S. Fan** [1] **and Ebede Ndi** [2]

1    Department of Industrial Engineering and Management, National Taipei University of Technology, Taipei 10608, Taiwan; morrisfan@mail.ntut.edu.tw
2    Xiantai Water & Electricity Company, Taipei 10608, Taiwan; ebedendi@gmail.com
*    Correspondence: feilungtw@gmail.com

**Abstract:** Fixing waterpipes using cryogenic technology is a complicated activity because it is difficult to see the transformation of water into ice inside a pipe. To solve this problem, a series of experiments and testing was carried out on three types of waterpipes—75, 100, and 150 mm cast-iron pipes (CIP), respectively—to monitor the external characteristics and the frost growth to accurately predict the completion of ice plug formation inside the pipe before applying the freeze-fixing method. A CCD camera was set up to capture four frosting images per minute and to send one image every 15 s to the computer for processing and for identifying the interface between frost and no frost. The results showed that when the brightness distribution along the pipe axis goes down from the 100% brightest and coldest region near the freezing jacket to 90% and hits the 1/10th mark away from the brightest area, which is the interface, the freezing process is finished, and the ice plug has completely formed and can effectively block water flow to allow safe repairs. In other words, the findings determined that the frost length was more or less equal to the ice plug length. Additional testing was done to increase water pressure up to 35 kg/cm$^2$ for about 10 min on one end of the freezing jacket, which is higher than the regular pressure testing of 10 kg/cm$^2$. When 0 kg/cm$^2$ was recorded on the other end of the freezing jacket, it was concluded that the ice plug was strong enough to resist high pressure. The success of these experiments and testing confirmed that this innovative freezing method was safe, cost-effective, and suitable to be used efficiently in semiconductor factories and modern buildings.

**Keywords:** hot tapping; frosting; ice plug; image recognition; pipe freezing; cryogenic technology

## 1. Introduction

As cities urbanized and grew, maintaining safe water supply systems became a necessity, including meeting requirements for public, commercial, and industrial activities both in quantity and quality. Risk assessment and risk management of water supply networks have also become a priority [1]. Delivering uninterrupted water supply from sources to customers entails the use of waterpipes. Yet, maintaining and fixing waterpipes has proven to be a delicate and sensitive activity as it involves public health and workers' safety. In 2016, researchers from Purdue University studied the release of toxic substances and hazards caused by cured-in-place pipe (CIPP) waterpipe repair sites [2]. CIPP is utilized for cesspool, storm drain, and potable waterpipe repairs and is the favored waterpipe repair technology applied in the United States; the toxic substances it transports are important risk factors for the environment and the nearby buildings [2]. Such an unsafe pipe-fixing method is evidence of the need to create a new and safe screening method that would minimize health risks and maximize cost-effectiveness.

Speaking of minimizing health risks and maximizing cost-effectiveness, studies have shown that damaged pipes lead to a large quantity of water waste in water distribution systems each day, destroying infrastructures and disrupting water supply [3]. For this reason, much work is being deployed to try to minimize leaks and improve the efficiency

and effectiveness of waterpipe functionality [4,5]. Other solutions have been to develop pipe failure models to proactively manage sound judgements and help identify pipes that need to be fixed or replaced. When a water supply network fails, it often results in a limitation or lack of water supply to the consumers [6]. Wasting water and losing money are the challenges that companies face throughout the world [7]. Faulty valves and damaged pipes can create major disruptions in intricate water distribution networks [8]. Our safe frost growth screening method is specifically designed to accurately predict the development of an ice plug inside a pipe, fix a failed pipe, and replace a faulty valve by isolating an area and installing a bypass to ensure an uninterrupted water supply. This means that we use LN2 to freeze the pipe and carefully monitor frost growth on either side of an attached jacket outside that pipe to accurately predict the complete development of the ice plug before we carry out repairs.

In a recent study, researchers looked at the correlation between the frost length and the pipe diameter [9], but in the current study, the authors look at the correlation between the frost length and the length of the ice plug inside the pipe. In this case, the frost length should be more or less equal to the length of the ice plug to accurately determine safe completion of ice plug formation. Earlier studies have also measured ice frost outside of tubes in an effort to ascertain the importance of monitoring frost growth in cooling systems [10].

In the pipe-freezing method, it is always difficult to accurately predict the complete solidification of the ice plug inside a pipe because first, we cannot see through the pipe, and second, even if we could see through, we could be distracted and/or not be able to predict with confident precision. Two methods have often been used to know when ice plug formation is complete: one is to look at the external features of the pipe [11], the other is to test water pressure [9]. The former is the method we have developed and recommend because it is the safest and most accurate prediction equipment in real time and in operation sites. The latter is risky because pressure rise can only be certain in laboratory observations; the pipe is very short and does not have enough space to tolerate the volume expansion which happens during water solidification. In actual operation sites, the pipe is long, depending on different cases, such that pressure rise can only be observed but we cannot be certain of the quantity or level of said pressure rise. However, pressure rise can be used as an auxiliary judgement but not as the sole prediction criterion.

Observation of frost formation requires consideration of surface roughness, meaning that the surface condition leads to an understanding of the pressure drop, heat transfer, and frosted heat exchanger [12]. Normally, due to the arbitrary distribution of nucleation sites, the very first layer of frost formation is likely to be nonuniform, and as a rule, the frost formed by crystallization from a moving vapor is of a porous, coarse structure, similar to the frost found in refrigerators although even more porous [12].

Studies have also measured the transient frosting coefficients of performance (COP) of cooling methods in relation to the time it takes for the frost to grow [13]. Other studies have looked at the connection between damaged pipe hazards frosting [14,15]. However, there is lack of research on measuring waterpipe ice expansion pressure on new plastic pipe frost heave [16]. Under working water pressure, the interplay between ice and the pipe wall leads to frosting [17]. Ice expands and generates pressure inside the pipe during water solidification [18,19]. It is not easy to measure the frosting process because its formation is complex [20]. Time and space are important factors when studying frost thickness and mass concentration [21]. Our method will use image processing equipment to circumvent the above-mentioned difficulty.

There are many studies on frost growth and its connection to pipe wall temperature [22–30]. All these studies have been very helpful in understanding phenomena such as condensation, dew point, heat transfer, dehumidification, subsaturation, supersaturation, and the interface during the frost formation process. Even though there is an expansive body of literature on frost growth [31], there is less focus on the mechanism of the frost

growth screening method as it helps to accurately predict ice plug formation completion inside a waterpipe during repairs.

In summary, methods for predicting frost growth trends and rate to understand the frost formation phenomenon have been proposed [32]. However, the scientific literature has mainly focused on studying the effect of frost on buried pipes or frost heave on the ground. Some have focused on oil pipes and others on fuel pipes. Very little research has studied frost growth methods as accurate predictors of ice plug formation completion during waterpipe freeze-fixing. This is the gap that the current study is trying to fill. The main question that drives this research is to find a way to accurately predict the completion of ice plug formation during freezing pipe repairs because of safety issues, especially in a clean room. In the absence of an accurate prediction method, the alternative would be to spend more time and liquid nitrogen to make sure the ice plug is formed completely. Our objective is to monitor frost growth characteristics on the pipe wall and gauge the ice forming process because of the correlation between frost length and ice plug length. To quantify frost length on the pipe wall, a real-time image identifier was used. With this innovative method, pipe fixing work can be carried out safely.

## 2. Materials and Methods

### 2.1. Experimental Procedure and Design

In our experiments, we used external devices on three types of waterpipes: 75, 100, and 150 mm cast-iron pipes (CIP), respectively, to monitor frost growth and accurately predict the completion of ice plug formation inside the pipe before carrying out repairs. Besides CIP, other materials such as steel or PE can also be used in water supply systems. However, our method can mostly be used on CIP and steel pipes with approximately the same results. But this method cannot be used on materials such as PE and PVC because the heat conduction is too slow and frost growth does not happen.

The theoretical basis or rationale for this method is based on experimental, historical, analytical, and observational evidence. We carried out several experiments over a couple of years using this method. Faced with the constant inability to accurately gauge the complete solidification of water inside the pipe before carrying out repair works, we decided to try a new method. We combined an image-processing CCD camera with a specifically programmed computer for data analysis to help us use the frost growth on the outer wall of the pipe to accomplish our task. The original contribution of this method is that it has never been used in the field of hot tapping and pipe freezing before. Below, we provide some context to better understand the theoretical and experimental foundation of our method.

In working on semiconductor bubble memory in the late 1960s, Willard Boyle and George E. Smith invented a charge-coupled device also known as CCD. That device was later used to reduce smear from bright light sources. Since we were looking for such a device, we then used it to develop a kind of image recognition flowchart, where a CCD camera would capture full color images and send them to the computer for grayscaling, binarization, and analysis. In computer-generated imagery, a grayscale image is one in which the value of each pixel is a single sample representing only an amount of light. During grayscaling, the computer measures the intensity of light at each pixel according to a particular weighted combination of frequencies. Although the grayscale can be computed through rational numbers, image pixels are usually quantized to store them as unsigned integers, to reduce the required storage and computation. After measuring the intensity of light, the computer begins the process of binarization. Binarization is the method of converting any grayscale image into a two tone black/white image. To perform the binarization process, first the computer finds the threshold value of grayscale and checks whether a pixel had a particular gray value or not. In our study, we were looking for the brightest area of the frost representing the completely solidified water in the pipe, so we programed the computer to receive images and convert them into numbers, using different color lines to help us measure the percentage of brightness of the frost growth. The CCD software (Frozen Line Recognition, version 002) does not detect the brightness of the frost

layer in the image. It only captures images and sends them to the computer for analysis. Following a specific program which is initially set in the computer, analysis can be made to determine the percentage of brightness of the frost layer as it relates to the formed ice plug inside the pipe. Figures 6 and 7 show in detail how the frost growth progression was calculated.

Our methodology follows a few relevant and connected steps. First, we present the modification of an old model that was published a couple of years earlier as we tried to improve our method. Then, we discuss the mechanism of heat transfer to help the reader understand the phenomena that happen inside the pipe during freezing. Next, we detail the frost formation process. Finally, we show how we use the CCD image processor to help us predict the completion of ice plug formation inside the pipe before we carry out repair works. The following Figure 1 below shows a water distribution network with a layout which is suitable to apply the freezing method during pipe fixing.

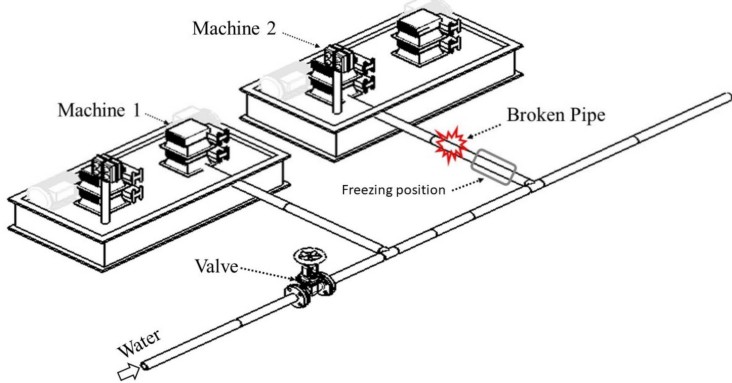

**Figure 1.** Broken pipe suitable for using freezing method.

### 2.2. Modification of the Old Model

In an earlier study, researchers examined the correlation between frost length and pipe diameter and found that the first was slightly longer than the second after ice plug removal [9]. Figure 2 shows the old model with a one-way LN2 outlet. T1 is the temperature during LN2 flowing in. T2, T3, T4, T5, and T7 are temperatures during vapor flowing out. T6 is the temperature of the freezing jacket. In that study, the objective was to observe the frost length in relation to the pipe diameter (D). The asymmetry between frost and jacket lengths presented a problem that needed to be corrected. In the current study, the objective is to measure the frost length in relation to the length of the ice plug inside the pipe. In this case, the frost length should be more or less equal to the length of the ice plug to accurately determine safe completion of ice plug formation. So, we corrected the asymmetry problem identified in the old model by modifying and redesigning a new model with a two-way LN2 outlet which ensures a near perfect symmetry between frost length and ice plug length, as illustrated in Figure 3 below.

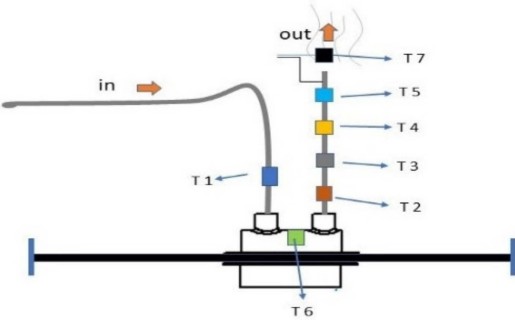

**Figure 2.** Old model with one-way LN2 outlet.

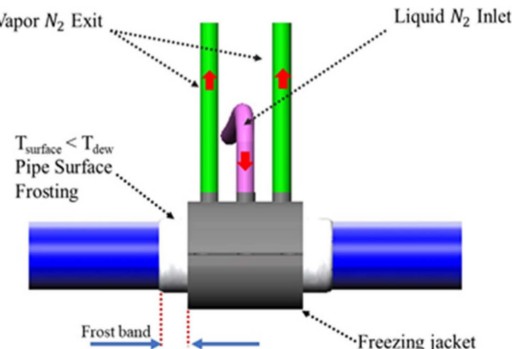

**Figure 3.** New model with two-way LN2 outlet.

*2.3. Heat Transfer Mechanism*

Figure 3 above shows the LN2 flow path. The liquid nitrogen flows into the middle pink inlet pipe shown in the red arrow at about −196 °C, absorbs heat from water inside the pipe through a mechanism called heat transfer, then evaporates through the two green outlets shown in red arrows into the environment, triggering a decrease in the water temperature to freezing level until the water turns into an ice plug. In a previous study, it was mentioned that this heat transfer mechanism was safe for the environment because of the fast vaporization effect of the released LN2 concentration even though there could be minor risks to the health of staff and equipment. To solve this problem, researchers installed sensors to detect potential changes in temperature as can be seen in Figure 2 above. They also calculated the speed of feeding LN2 to avoid disruption in heat transfer during water solidification.

Figure 4 is an illustration of the heat transfer mechanism inside a waterpipe. Zone A shows a two-phase flow, Zone B heat conduction, and Zone C natural convection. During violent heat transfer, as LN2 begins to absorb heat from water, bubbles are formed. A prior study looked at bubble formation and discovered that heat flux was a key element of heat transfer [33]. So, as water begins to turn into an ice plug, the volume expands at from 7 to 8% exerting a high pressure inside the pipe, and because its inner surface is rough, water and later ice fills its cavities. The roughness of the pipe wall helps sustain maximum pressure and prevents the ice plug from shooting out like a bullet. The mechanisms produced during heat transfer, namely volume expansion, friction, and filling up pipe wall cavities, have helped to improve the freezing method. Takefuj and Okubo [34] examined the friction between the waterpipe and the frozen ice plug; they changed the globally widely used single-ice plug freezing method and proposed a double-ice plug freezing method with the following advantages: (a) significant increase in water pressure in order to stop water flow, (b) frozen time is the same for both single-ice plug freezing and double-ice plug freezing, (c) the ice adhesive strength or the ice friction of the double-ice plug is roughly four times higher than that of the single-ice plug, and (d) the double-ice plug freezing requires less liquid nitrogen than the single-ice plug.

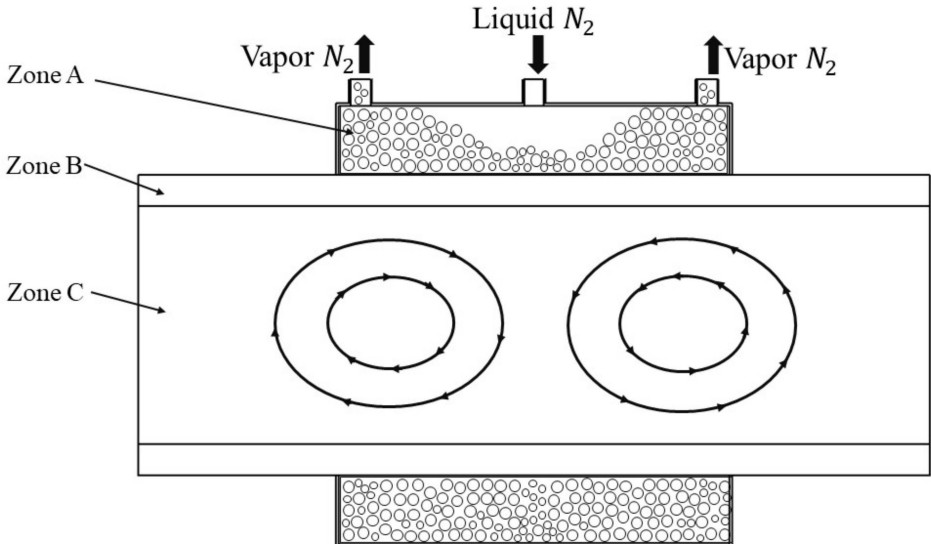

**Figure 4.** Illustration of heat transfer mechanism.

*2.4. Frosting Process*

Monitoring frost growth to accurately predict ice plug formation requires that we understand what is happening inside the pipe, such as the heat transfer mechanism and how and when water turns into ice. At the same time, we must also understand the mechanism that happens outside the pipe and how that affects frost growth.

Three conditions must happen to ensure the formation of frost: surrounding relative humidity (RH), surrounding temperature (T), and dew point (DP). Surrounding RH and surrounding T tell us how much water vapor there is in the surrounding air. If there is no water vapor in the air, there will be no frost. DP is the deciding factor of whether or not water vapor will condense. These three conditions are important because we are trying to answer the question: how low should the temperature be for frost to form? RH is relative and not absolute because it is compared to the maximum water vapor pressure. The maximum amount of saturated water pressure is not constant. For example, at 25 °C, saturated water pressure should be 3.1698 kPa. If water vapor is 2 kPa, RH should be 63%. That means if the real water pressure is 2.000 kPa and the maximum amount of saturated water vapor pressure is 3.1698 kPa, when we divide the former by the latter at 25 °C, we obtain 63% RH. The real water vapor pressure in a clean room should be 1.9019 kPa. This creates another condition: how low should the temperature be for water vapor to condense? If the temperature is low enough, water vapor will condense to form liquid water. Similarly, if the pipe wall temperature is lower than the water frozen temperature, frost will form. In the same vein, when the temperature of the waterpipe wall is lower than the water frozen temperature, we can guarantee that frost will form. Generally, if RH is very high, DP will not be too far from the surrounding temperature. For example, at 25 °C, if RH is 95%, DP could be 24 °C. That means if T < 24 °C, there will be water droplets, and if T < 0 °C, there will be frost. Once we integrated all the atmospheric conditions and make sure that frost will form, we installed a CCD camera to monitor frost growth and range.

*2.5. CCD Image Processor*

We used a CCD camera to monitor frost formation and range. This method helped us to see how long the frost would be in relation to ice plug length. Our experiments showed us that this external equipment is more reliable than using our bare eyes. The CCD camera gets four frosting images per minute and sends one picture per 15 s to the computer for processing. Those images were then analyzed to gauge the state of ice plug formation inside the pipe and the frost characteristics outside the pipe. Using frost brightness variation, CCD-processed images allowed us to determine the interface between the near and far

regions of the frosting area. Figure 5 shows how the CCD camera equipment was set up with a focus on the frosting area and a connection to the computer.

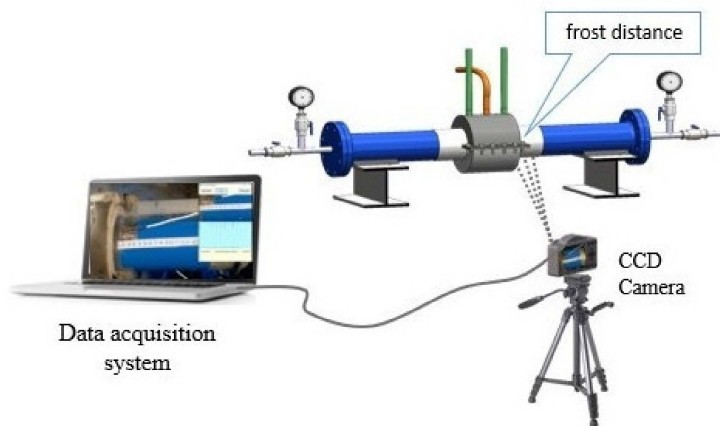

**Figure 5.** CCD image processor.

## 3. Results and Discussions

The experiments were conducted under the surrounding temperature of 26 °C and relative humidity of 60%, which were similar to a clean room environment. By simple calculation using a thermodynamics steam table, the dew point temperature was about 17.6 °C. In Figure 6, we see that when we poured LN2 into the freezing jacket, the temperature on the surface of the pipe near the jacket decreased very quickly and became lower than the dew point temperature. This meant that small water droplets would soon condense on the pipe surface. Then, the pipe surface temperature would also be lower than the water frozen temperature because of the good thermal diffusivity of the pipe wall. Finally, the condensed water turned into frost and gradually covered the area along the pipe axis. In this experiment, the pipe wall and the water were both under low temperature. But the question could be: why does the temperature of the pipe wall go down faster and why is it colder than water? That is because when we fed LN2 into the jacket, the pipe wall, having large thermal diffusivity, could pass the cold temperature very fast since the pipe wall temperature takes little heat; the temperature would then drop one degree, rapidly increasing conductivity. That is also why the range of cold temperatures is wide on the pipe axis. However, if we kept pouring LN2 in, the water inside the pipe would decrease very slowly because thermal diffusivity became small. So, the temperature would go down very slowly, heat transfer would happen very slowly, and it would take a long time to turn water into an ice plug. To prevent this from happening, we mentioned an earlier experiment that recommended injecting the right amount of liquid nitrogen into the inlet and at a proper speed, between 0.7 and 0.8 kg/min, to ensure a smooth heat transfer mechanism followed by a safe completion of the ice plug formation process [9].

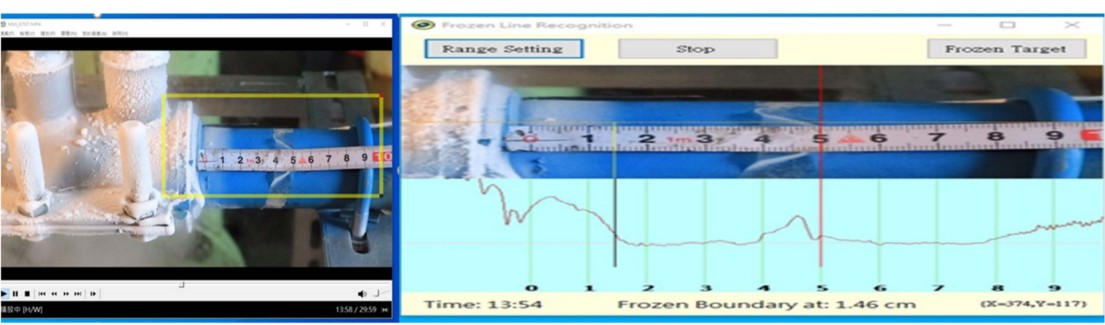

**Figure 6.** Frosting interface.

So, CCD-processed images calculated the brightness on the pipe axis and determined the interface. We compared the brightness distribution. The brightest area around the jacket where we injected LN2 would be normalized to avoid decimal numbers and rounded into the integer 1 to facilitate the calculation operation. We used 1 to indicate the 100% brightest area because that was where it became cold very fast. Then, along the pipe axis the brightness went down. If the brightness went down 90%, we determined that this was the interface between frost and no frost, which meant that the brightness was only 1/10th of the brightest area, and that gave us enough data to predict the length of the ice plug inside the pipe. When we injected LN2 through the jacket, the cold temperature went through the pipe wall. First, we chose the area to connect with the CCD software for analysis. We did not need to monitor a large area. We chose the range that we needed to monitor (shown in the yellow frame on the left side of Figure 6) because we knew that the chosen area was larger than the frosting limit, which depended on the pipe diameter. Within this area, when the frost length went over the limit, the software showed a "job finished" message on the computer screen. During the frosting process, a ruler should be used to monitor the length of the frosting area with the software-identified scales to determine the real-time frost length. We also chose the baseline shown on the right side of Figure 6 to calculate the degree of brightness spread along the axis of the external pipe wall because we knew that the frost length was more or less symmetrical to the ice plug length. The real-time frosting interface (vertical green line) is located at the first position where the brightness was only 10% of the brightest area (see yellow-framed monitoring area on the left side of Figure 6). The vertical red line on the right side of Figure 6 is the frosting limit set before injecting LN2 into the jacket.

Figure 7 shows the experiments on the waterpipe's external characteristics. CCD image processing can take as many images as we like, but because the frosting process is slow, we set the system to capture only four frames per minute, one every 15 s. In Figure 7, we chose three images to analyze the pipe surface frosting range captured at (a) 15 s, (b) 13 min 30 s, and (c) 21 min 20 s after starting the freezing process. Earlier, we said that if the brightness of a captured frame somewhere on the pipe axis is 10% from the brightest area, we call that area the interface. So, every 15 s, the computer would analyze the brightness distribution, moving away from the brightest area near the jacket as the frost expands. This helped us to decide where to stop the freezing process. In a previous study, it was shown that the length of the ice plug depended on the length of the freezing jacket, and because the length of the ice plug was 1.5 times the length of the jacket diameter, once we reached that ice plug length, it was good enough to stop freezing [9]. In this experiment, we were trying to show that the length of the ice plug inside the pipe was more or less equal to the length of the frost outside the pipe, and based on our experience, finding 1/10th brightness away from the brightest area in the CCD-processed images was good enough to stop freezing as it almost accurately predicted the length of the ice plug. It is worth noting that even if we changed the baseline number, it would not make much difference.

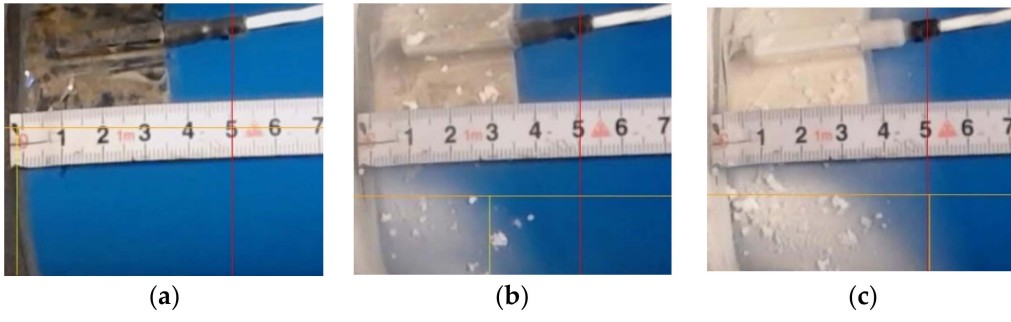

(**a**)                  (**b**)                  (**c**)

**Figure 7.** Pipe surface instant frosting interface position captured at (**a**) beginning; (**b**) 30 min and 30 s after starting the freezing process; (**c**) ending; (pipe diameter 75 mm).

After injecting LN2 into the inlet, in the first image Figure 7a, the CCD camera could not find any sign of frost on the external pipe wall. The real-time frosting progress indicator is at the zero-mark of the ruler (see yellow line of frame a). The experiment continued and after about five minutes, frost began to form. The brightness also began to expand along the pipe axis. In Figure 7b, the CCD camera sent a brightness distribution image to the computer, which then calculated the percentage of the brightness expansion trying to determine the interface between frost and no frost. We randomly chose image (b) at 13 min 30 s. We could have chosen any image at any timeframe without a significant difference. The important element was to accurately determine the interface. The temperature of the pipe surface near the jacket began to decrease rapidly and became lower than the dew point temperature, which meant that small water droplets would condense on the pipe surface and will quickly turn into frost. The good thermal diffusivity of the pipe wall made the surface temperature lower than the water frozen temperature. The condensed water droplets turned into frost and gradually covered the area along the pipe axis. At this point, the frosting interface indictor was at about 3 cm from the origin. Image (c) shows the real-time frosting interface position at 5 cm, which is just across the frosting limit. The computer sent a "job finished" message. This meant that the ice plug inside the pipe had completely formed. The length of the ice plug could effectively stop the pressurized water, and repairs could now begin.

Figure 8 shows the length comparison between the frost and ice plug of the pipe diameter 75 mm. Before starting the freezing process, the pipe pressure was the same as in the factory. The initial condition was about 2 or 3 $kg/cm^2$. When we finished the experiment, the pressure went high because of the ice plug expansion inside the pipe. Because the pressure increase was not stable but depended on different conditions, we conducted pressure testing to make sure that the ice plug could resist high pressure and any force that was exerted on it. Thus, we had the equation $P \times A = F$ (pressure multiplied by area equals force). On the left side of the jacket, we increased pressure for up to 35 $kg/cm^2$ for about 10 min, and on the right side, we had 0 $kg/cm^2$. We then observed that the ice plug could effectively and efficiently withstand the 35 $kg/cm^2$ pressure difference. At the end of testing, the freezing jacket was removed so that we could measure the total length of the frosting area, which was about 385 mm. In general, the ice plug could stay firm in position for a long time without the freezing jacket. But in our study, we needed to take out the ice plug and measure its length as soon as possible, then open running water on the pipe surface for about 2 min. The fast heat convection helped to melt down the ice in the contact area between ice plug and inner pipe wall. This allowed us to take out the ice plug quickly without affecting its length. The shape of both ends of the ice plug were concave in the centre, which meant that the length of the ice plug in contact with the pipe wall was longer than the length in the centre. This phenomenon is caused by the natural convection of water during freezing since the pipe wall temperature was always lower than the temperature in the pipe centre. Contrary to the argument that suggests that we should measure the shorter inner side of the ice plug, not the longer side, to understand the pressure-resistance phenomenon and ensure safety, we demonstrated through our testing that it was only when there was contact with the pipe wall that it could resist the pressure difference. The longer size of the ice plug at the contact area with the pipe's inner wall helped to keep it steady during pressure testing because the shear stress between the ice plug and the pipe's inner wall was the reaction stress to resist the force created by the pressure difference between the two ends of the ice plug. In other words, the normal pressure of the area should be equal to the sheer pressure multiplied by the Area. Thus, we had $P \times A_n = \sigma \times A_s$. A larger contact area between ice plug and pipe wall meant that the ice plug could resist a larger pressure difference during pressure testing. The length of the ice plug shown in Figure 8 is about 400 mm, which is close enough or more or less symmetrical to the frost length. One important problem with freezing fixing work is that it was hard to make sure whether the ice plug has completely formed. This is also related to safety issues. By monitoring the outside characteristics of the pipe wall, such as frost

growth and pressure testing, the effective completion of ice plug formation can ensure safe repair works.

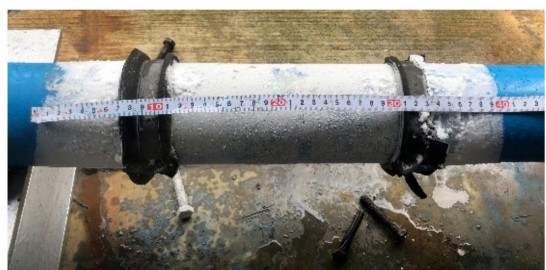 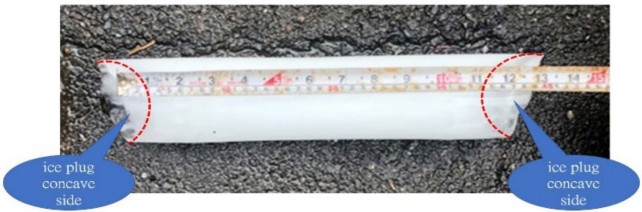

**Figure 8.** Length comparison between frosting characteristic and ice plug.

Figure 9 shows water pressure gauge readings at both sides of the jacket during pressure testing. Before starting the freezing process, the initial pressures of both gauges were 2.5 kg/cm$^2$, which was similar to the factory water supply system pressure. During freezing, water inside of the pipe gradually began to turn into an ice plug; and since ice density is smaller than water, the pressure inside of the pipe also progressively increased. In this study, the experimental pipe was only 1.5 m long, such that there was not enough room to tolerate the volume expansion as both sides of pressure increased. The amount of increased pressure depends on the pipe diameter. For the 75 mm small pipe, the pressure can increase up to 30 kg/cm$^2$ at the end of freezing. For the 150 mm large pipe, the pressure can increase up to 12 kg/cm$^2$. But in the factory water supply system, the pipe is longer and maybe include storage tank facility, which renders the predictability of the pressure increase obsolete. That is why we ran pressure testing before undertaking repair works to mitigate any unpredictable eventuality. The pressure rising during freezing could still be observed, but it was difficult to estimate the amount. In order to make sure that the ice plug could effectively stop water at high pressure in different conditions, 10 min of pressure testing was applied to all the experiments. Water pressure at one side of the jacket was increased up to 35 kg/cm$^2$, which is higher than regular pressure testing of 10 kg/cm$^2$. The other end was opened, and the gauge reading was 0 kg/cm$^2$, denoting just the atmospheric pressure. During pressure testing, no amount of water flowed through the open end, showing that the ice plug had effectively stopped water flow under high pressure.

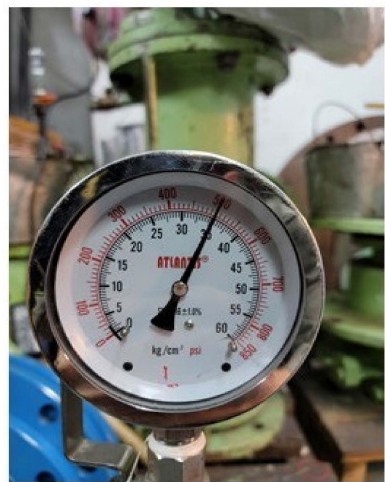 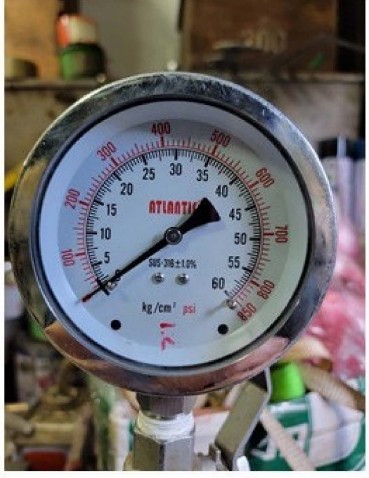

**Figure 9.** Water pressure gauge readings at both sides of jacket during pressure testing.

The results of three pipe nominal diameter dimensions are listed in Table 1. The testing pipe lengths were all the same—1500 mm long—and the freezing jacket length was 250 mm

for the 75 mm diameter pipe. The larger diameter 150 mm pipe needed a longer ice plug to withstand the pressure testing, such that a 300 mm freezing jacket should be used. The results show that when the pipe diameter is larger, the freezing process takes more time. The frosting range is also wider and the length of the ice plugs are longer. The absolute relative errors between frost and ice length are under 4%, which means that the time to stop the freezing process can be determined scientifically and safely. However, for the other bigger diameter pipes, the natural convection effect is more serious. The topics of frosting and ice plug symmetry and the long freezing time problem still need further study.

**Table 1.** Comparisons of frost and ice plug length.

| Pipe Nomial Diameter Dimenion (mm) | Freezing Jacket Length (mm) | Freezing Time (minute) | Frost Length (mm) | Ice Length (mm) | Frost Ice Relative Absolute Error (%) |
|---|---|---|---|---|---|
| 75 | 250 | 21.25 | 385 | 400 | 3.9 |
| 100 | 200 | 45.5 | 310 | 320 | 3.2 |
| 150 | 300 | 60.25 | 470 | 480 | 2.1 |

## 4. Conclusions

The goal of this study was to develop a safe frost growth screening method, which was specifically designed to accurately predict the completion of ice plug formation inside the pipe. The main question that drove this research was to find a suitable way to confidently gauge the completion of ice plug formation during freezing pipe repairs. Our objective was to monitor frost growth characteristics on the outer pipe wall and assess the ice forming process because of the correlation between the frost length and the ice plug formation completion.

In order to quantify the frost length on the pipe wall, a real-time image identifier was used. The CCD camera was set up to capture only four frames per minute, one every 15 s. Those images were then sent to the connected computer for processing and analysis. The results of the analysis showed that the frost length is more or less the same as the ice plug length. This method can be effectively applied to the work of pipe repair. However, a few limitations to this study are the fact that the CCD camera cannot identify the brightness distribution when the outer surface of the pipe is bright or white before initiating the freezing process. Also, frosting presents more challenges when we use PVC pipes. At the same time, interesting future research could focus on developing a CCD capability that can monitor temperature and water pressure combined with frost length, a kind of integrated technology with the objective of improving safety and reliability. Another promising item of future research could be to look at the correlation between temperature difference, gravity, diameter size, and natural convection.

Carrying out repair works on waterpipes without shutting down the entire water supply system, recording the freezing time, securing the safety to withstand high water pressure during repairs and maintenance, and being able to predict the completion of ice plug formation more accurately inside the pipe all require continued improvement in water freeze-fixing methods. This study contributes to the amelioration of safe gauging and monitoring methods and equipment in the field of pipe freeze-fixing.

## 5. Patents

Liu, F. L; Chiang, T. Y. Chiang. Application of a frozen ice blocked pipe water method to non-shutdown repairing work of water supply system. Patent No. TW201938923A.

**Author Contributions:** Conceptualization, F.-L.L. and S.-K.S.F.; methodology, F.-L.L. and S.-K.S.F.; formal analysis, F.-L.L. and S.-K.S.F.; investigation, F.-L.L. and E.N.; resources, F.-L.L.; data curation, F.-L.L.; writing—original draft preparation, E.N.; writing—review and editing, E.N.; project adminis- tration, F.-L.L.; supervision, F.-L.L.; funding acquisition, F.-L.L. All authors have read and agreed to the published version of the manuscript.

**Funding:** The authors declare that this study received funding from ST Water Company. The funder was not involved in the study design, collection, analysis, interpretation of data, the writing of this article or the decision to submit it for publication.

**Data Availability Statement:** The data presented in this study are available on request from the corresponding author.

**Acknowledgments:** Special thanks to all those who supported the authors in this research.

**Conflicts of Interest:** Author Ebede Ndi was employed by the company Xiantai Water & Electricity Company. The remaining authors declare that the research was conducted in the absence of any commercial or financial relationships that could be construed as a potential conflict of interest.

## Nomenclature

| | |
|---|---|
| A | area |
| An | area of ice plug cross section |
| As | area of ice plug in contact with the pipe wall |
| kg | kilogram (basic unit of mass) |
| cm | centimeter (unit of displacement) |
| T | temperature (degree of heat present in a substance or object) |
| °C | degree Celsius (unit used to measure temperature) |
| min | minute |
| sec | second |
| t | time |
| $cm^2$ | square centimeter |
| mm | millimeter |
| ~ | approximately |
| < | less than |
| σ | sheer pressure |
| D | diameter |
| P | pressure |
| F | force |
| LN2 | liquid nitrogen |
| CIP | cast-iron pipe |
| CCD | charge-coupled device |
| CIPP | cured-in-place pipe |
| COP | coefficients of performance |
| HF | heat flux |
| HTC | heat transfer coefficient |
| RH | relative humidity |
| DP | dew point |
| kPa | kilopascal (one thousand times the unit of pressure and stress in the meter–kilogram–second system) |
| PVC pipe | polyvinyl chloride pipe |

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
