# Peer review of "A Safe Frost Growth Screening Method to Accurately Predict Ice Plug Formation Completion during Waterpipe Freezing Repairs"

_water, doi:10.3390/w16020202_

Round 1

Reviewer 1 Report

Comments and Suggestions for Authors

The article concerns the water pipes repairs. It may be interesting for readers of Water. In general, this manuscript is well organized and written. The following requests/suggestions should be taken into account to improve the quality of the manuscript.

-        Is the proposed method suitable for water supply networks? I understand that the pipe must be dug out first? Is it more about internal water systems?

-        CIP is one of the materials used. Other materials are often used in water supply systems, such as steel or PE. Please refer to the possibility of using this method on these materials.

-        The research concerns small diameters. What about large diameter pipes whose failures are associated with the most serious leaks?

-        In practice, it is not necessary to shut down the entire water supply system after a failure (line 435). There are many gate valves installed in the system that enable cutting off a selected part of the water supply system.

-        In the event of a failure, it is necessary to quickly close the flow to limit water leakage. Is the proposed method faster than closing the valve?

-        Is the use of the proposed method economically justified? Is the method cost-effective compared to installing additional gate valves?

Author Response

Thank you very much for taking the time to review this manuscript. Please find the detailed responses attached to the word document that has been uploaded.

Reviewer 2 Report

Comments and Suggestions for Authors

The main contribution of this manuscript is to develop a safe method of screening frost growth, which utilizes the principle that the length of the frost layer is roughly the same as the length of the ice plug formed, as proven by experiments. By using CCD imaging to observe the brightness of the frost layer in real time, the length of the frost layer can be determined, and thus the formation of the ice plug inside the pipe can be accurately predicted. The authors claim that this method can be effectively applied to the work of pipe repair. However, there are some areas that need to be improved in this manuscript, and I suggest that the author should revise it before publication.

1. This manuscript performed ice plug formation experiments on cast iron water pipes with three different diameters, modified the previous experimental model, and sought to establish a relationship between the length of the frost layer and the formation of ice plug, so as to measure the length of the frost layer to estimate the formation of ice plug. I think this research has limited contribution to the field, suffers from a lack of innovation and theoretical depth, and the research conclusions are preliminary.

2. The authors mentioned that they observed the brightness of the frost layer to determine its length: “If the brightness decreases by 90%, we infer that this is the boundary between the frost-covered and frost-free regions, which implies that the brightness is only 10% of the maximum value.” What is the theoretical basis for this method? The authors should clarify this point.

3. How does the CCD software detect the brightness of the frost layer in the image? What is the methodology behind it? The authors should explain this in detail and present the output of the CCD software. This would enhance the completeness of the experimental process and the validity of the results.

4. There are some formatting errors in the manuscript, such as extra spaces in the sentences of line 197 and 242.

5. The language of the manuscript is not concise enough. For example, there are excessive and lengthy descriptions in the introduction and section 2.4. The authors may consider using more figures and tables to illustrate their points.

Author Response

Thank you very much for taking the time to review this manuscript. Please find attached the detailed responses in the word document that has been uploaded. 

Round 2

Reviewer 1 Report

Comments and Suggestions for Authors

The manuscript has been improved as requested.  

Reviewer 2 Report

Comments and Suggestions for Authors

The review comments have been addressed. This manuscript can be considered acceptance.